# “Just Be Strong, You Will Get through It” a Qualitative Study of Young Migrants’ Experiences of Settling in New Zealand

**DOI:** 10.3390/ijerph18031292

**Published:** 2021-02-01

**Authors:** Enoka De Jacolyn, Karolina Stasiak, Judith McCool

**Affiliations:** 1Faculty of Medical and Health Science, University of Auckland, Auckland 1142, New Zealand; 2Department of Psychological Medicine, Faculty of Medical and Health Science, University of Auckland, Auckland 1142, New Zealand; k.stasiak@auckland.ac.nz; 3Epidemiology and Biostatistics, School of Population Health, Faculty of Medical and Health Science, University of Auckland, Auckland 1142, New Zealand; j.mccool@auckland.ac.nz

**Keywords:** migration, migrant, adolescent, adaptation, resilience, acculturation, New Zealand

## Abstract

Migration, when it occurs during adolescence, is particularly challenging as it coincides with a myriad of other developmental and social changes. The present study set out to explore recent young migrants’ experiences of settling in New Zealand. The qualitative study aimed to identify areas of particular challenge, examples of resilience and new insights into the acculturation process. Focus group interviews were conducted with migrant youth aged 16–19 from three urban secondary schools in Auckland The interviews were audio-recorded, transcribed and analyzed using a general inductive method. Key themes centered on new beginnings, confronting new realities, acceptance, support seeking and overcoming challenges. Young migrants in this study shared similar challenges during the early post-migration period. They were often faced with additional responsibility, being caught between two cultures while struggling with communication and language. However, they were able to draw on their own self-growth, gratitude, and social connections. This study provides an insight into experiences of young migrants in New Zealand, and offers suggestions for developing culturally relevant support to foster migrant youth wellbeing.

## 1. Introduction

Many migrants face a variety of acculturation stressors during the migration journey. Previous research suggests that learning a new language, finding new housing or employment and potentially being isolated from their extended family and familiar culture are among the most pervasive concerns [1]. The process of migration may be an additional stressor for migrant youth, in addition to those synonymous with adolescent social and emotional development. Developing a sense of self can become a challenge due to the pressure of having to negotiate and engage in the behaviors and values consistent with their own cultural background as well as those of their host country. Young migrants who often must maintain tradition values and beliefs at home, in school, are expected to fit in with their peers of the host country’s dominant culture. Generation 1.5′ is a term to describe youth who undergo a process of migration during their late childhood or early adolescence [2]. Generation 1.5 is also of interest because of their position of convergence in many social forces; for instance, they are situated between their countries of origin and residence, they are between childhood and adulthood, they are often between their parents and the host society, and in New Zealand, they are between Pākeha majority and Māori (indigenous peoples). Furthermore, Generation 1.5 navigating New Zealand, an already established as a multi-cultural and ethnically diverse country [3]. Recent work by Marlowe [4] explores how young migrants maintain established networks while forging new relationships with the support of social network tools. Interest in the concept of resiliency here is channeled toward an interest in connectivity—both actual and virtual [4].

Young or adolescent migrants can be overlooked in research with the tendency to focus on refugee and adult migrants’ experiences. New Zealand-specific literature about the needs of young migrants is particularly scarce despite the rapidly growing migrant population [5,6,7]. Over a quarter of New Zealand’s five-million people were born overseas [8] putting it in the top five most ethnically diverse countries within the OECD [9]. Auckland is the largest city comprising of over 1.5 million residents and it has the highest proportion of migrants with almost two of every five migrants (39.1%) living there. Approximately 53% of Aucklanders are European, 27% Asian, 17% Pacific and 12% Māori (indigenous people) [10]. The recent growth in migration has created a debate on costs and benefits for both the host country and the migrants themselves [11] and how the emerging multiculturalism may affect the power sharing relationship between Māori and NZ Europeans, potentially threatening the indigenous status and rights of Māori [12]. Existing research suggests that most New Zealanders generally have a positive attitude toward migrants [13] because they increase skills, innovation, add cultural diversity and contribute to creating productive, vibrant, and healthy communities [14]. On the other hand, some surveys have revealed that Auckland residents viewed migrants more negatively than residents of other cities citing the changing of the city’s culture, loss of employment opportunities and increased crime [15]. Importantly, youth (aged 12–24) make up almost a quarter of the migrant population according to the latest census [10]. More than half of youth migrants who have migrated in the past five years live in the Auckland region, with China and India being the main source countries of recent young migrants.

Methods of research among young migrants have evolved to encompass a wider range of approaches that reflect the lived experience of young migrants. As young people hold an important place in society and are often touted as the ‘future leaders’; accordingly, the voices of this group deserve to be heard. The lived experience of young migrants is key to building an understanding of young migrants’ resilience and strategies used for adaptation to a new environment [16,17]. The aim of this study was to understand the lived experiences, and expectations, of being a young migrant in New Zealand. This information provides real time evidence to support the development of effective, appropriate support services.

## 2. Methods

### 2.1. Design

A qualitative study design, using semi-structured interviews, were conducted to generate textual data for a thematic analysis. This study design provided rich descriptive narrative data on young migrant’s experiences of settling into New Zealand. This approach also facilitated an understanding of young people’s experiences of migration, the distinct and uniquely adolescent experience, and how this group make sense of this period of transition [18].

### 2.2. Sample

A purposive, convenient sampling method was used to recruit students from Auckland high schools according to the following inclusion criteria: Participants over 16 years (i.e., legally able to consent for themselves); attending a secondary school within the Auckland Region (largest city in New Zealand, population approximately 1.5 million); migrated to New Zealand within the last 5 years; able to speak conversational English.

### 2.3. Procedure 

Advertisements were posted around schools identified as hosting a high proportion of migrant students. Students interested in participating in the focus groups contacted the researcher who arranged for them to attend one of the 40-min focus groups held in classrooms during the lunch break. Gender specific focus groups were conducted in a comfortable and private space to enable participants to express their thought and opinions in a safe location. All participants completed an informed consent form and were reminded of the voluntary and confidential nature of the interviews. 

All focus groups were facilitated, transcribed and initially coded by EDJ who is a female former migrant. Prior to the focus groups, participants were given the opportunity to ask questions about any aspect of the study including ethical considerations and guidelines of being part of the focus group. Each participant then completed a demographic questionnaire and were reminded of the ethical considerations. Refreshments (kai) were then shared to demonstrate respect for the participants volunteered time. During the focus group, a semi-structured interview guideline was used to guide a collaborative discussion of the youth’s perspectives and views. The interview guidelines included the following topics: re-settlement (e.g., initial thoughts and feeling about moving to New Zealand); wellbeing (common beliefs about wellbeing); support tools (where did you find support? How did social media play a role?). Interviews were audio-recorded for transcription and subsequent textual analysis. 

The research was conducted following the confirmation of ethical approval from the University of Auckland Human Participants Ethics Committee. 

### 2.4. Data Analysis

A General Inductive Approach (GIA), was used to analyze the qualitative data. GIA involves systematically organizing and interpreting the data collected and using categories and themes to identity patterns within the information [19]. Following the verbatim transcription of the focus group interviews, the textual data was then imported into NVivo 11 for initial open coding. These codes which were primarily descriptive, were subsequently refined by collapsing and merging initial codes into the final set of themes that are presented as results.

## 3. Results

This chapter presents a thematic analysis of the data gathered from the collective voices of the young migrants who took part in the focus group discussions. A description of each theme is supported by verbatim quotes from a participant present evidence and nuanced account of the participants’ experiences and the analysis conducted by the team.

In total, 28 participants aged from 16 to 19 years took part in the study (Table 1). All participants were first-generation migrants, meaning they were born outside New Zealand and have migrated to New Zealand within the past five years. The participants represented 11 nationalities, mostly from the regions of Asia but some also came from Oceania and South America. On average, participants had been living in New Zealand for approximately two and a half years and five of the 28 participants had been in New Zealand less than one year. Most participants migrated with their parents and continue to live with them. Participants belonged to three different schools within Auckland with decile ratings of 7, 6 and 4. The decile rating indicates the socioeconomic position of the school’s community in relation to other schools in the country with decile 1 schools with the highest proportion of students from low socioeconomic communities [20].

### 3.1. New Beginnings

Most participants in this study described feeling apprehensive and uncertain about leaving their familiar surroundings; friends, family, school, food, culture and climate. However, they also felt it was a ‘worthwhile’ decision which created ‘mixed feelings’. Participants spoke of looking forward to the new beginning mainly due to their countries of birth lacking social services that their families valued; a good education system with a pathway to a more prosperous future. Migrating meant a sense of hope for improvement in their quality of lives, many of the young people were eager to try something new. Despite the promise of new prospects, participants expressed a range of initial reactions to migrating. 

*I was in the middle, I was both happy and sad ‘cause I got to move overseas and meet new people but at the same time I didn’t know how I was going to meet new people and fit in*.—*Female, 17 years old, Sri Lanka*

The sense of excitement about moving to New Zealand was influenced by the degree of involvement the young person had in the decision to migrate. Those from families who involved them in the decision, were more likely to feel motivated to do well in school, which may have been due to a greater sense of independence. Having the autonomy to make decisions at this particular time of the participants’ lives appeared to be important. Being part of the choice of migrating helped to determine how these young people perceived their migration story. 

*If you coming to New Zealand and you have a goal in your mind, so I mean you just come here for that goal and then try to achieve it so it’s probably a good thing. But I don’t know, maybe someone might be forced to go to New Zealand and just that feeling of being forced, they might not enjoy. Even, like if you had a goal but you might struggle still ‘cause you don’t really care about it*. —*Male, 19 years old, Iran*

The majority, young people in this study explained that they held on to preconceived ideas about how they would fit into their new society. Their mixed feelings about migrating often included doubts about fitting in, making friends and being accepted. These worries were rooted in their own experiences up growing up in their former countries and were thought to be reflected again in New Zealand. However, participants tended to reflect on their migration journey positively, although this journey was not always easy. 

*For me, like even in Australia, a lot of people weren’t nice people so when I came here, I was maybe thinking oh maybe I will get bullied because they think I’m different. But it’s like, people are nice here, it’s strange but people are strangely nice here and I don’t know why that is, but I like it*.—*Female, 16 years old, Australia*

### 3.2. Confronting the New Reality

An inevitable part of migrating to New Zealand meant facing various challenges such as feelings of isolation, negotiating new roles and identities, various challenges in school, language barriers and cultural differences. Participants felt that facing these challenges set them apart both from other New Zealand born youth and their friends back in their former countries. 

*Just missing home. I think the biggest part for most of us is… when I first come here, the first year I was really missing home, the friends because at the time, I don’t have much friend. And um… I am quite poor, the English is poor, and everything is just poor. Quite bring me down. Just don’t feeling well*.—*Male, 16 years old, China*

The most challenging part of migrating was that they missed their old friends back in their former countries in addition to not having friends on arriving in New Zealand. For most, this contributed to the feeling of isolation and disconnection. All participants in this study emphasized the importance of establishing good friendships in order to feel settled in New Zealand. Perhaps to counteract this loss of old friends, the young people interviewed indicated that being surrounded by family acted as a buffer to protect their emotional wellbeing.

*I came here with my family so I am not really scared but when I am at school and I not know any English I would be like really scared*.—*Female, 17 years old, Cambodia*

In addition to feelings of isolation, some participants emphasized the challenges associated with taking on more responsibility when they migrated to New Zealand. A common example of this was when young people had learned English much faster that their caregivers and thus were responsible for translating or taking ownership of things that normally are attended to by adults. 

*I learnt English faster, its five or six times better than my mums… like I had to be at the bank with my mum and I had to talk for her like a translator, or like a call from Vodafone or Telecom*. —*Male, 18 years old, Sri Lanka*

*I think different for me because the way I learn English is different to how my parents learn English… they constantly need my help when they are speaking to someone else… Mostly I take the responsibility for it, like car insurance, things that someone back home might not do, it might be more useful… I never thought I could be doing things like this. It makes me prepared for a new life*.—*Male, 19 years old, Iran*

Perhaps unsurprisingly, most participants articulated how they initially struggled with English proficiency. The language barrier affected many areas of their life. For instance, it impacted their communication and interaction with their peers and teachers at school as well as their ability to learn.

*What happened was like, in my math class, my teacher was asking me to answer some questions. I know how to do it, but I don’t know how to say it, so I just say I don’t know, it happened like a lot of times, not just one*. —*Male, 18 years old, Sri Lanka*

### 3.3. Acceptance

Despite the many challenges faced by young migrants arriving and settling in New Zealand, many expressed a clear sense of appreciation and gratitude among the participants for living in New Zealand. Most participants agreed that the sense of safety, New Zealand’s lifestyle, better education, and the ability to gain invaluable experiences as being the top factors. 

*When I think about my future…like I wanna live in New Zealand… if I go to uni and study after I get a good job, I will be happy. I feel sad when I think about my past because maybe if I was back in India, I couldn’t get a better job. So now when I think about my future, I feel happy*.—*Female, 17 years old, India*

Throughout the discussions, participants expressed how the diverse society in New Zealand allowed them to acquire a deeper understanding of different cultures and meet people with different background. New Zealand’s culturally diverse society was seen positively by the young migrants, in particular, the diversity of religion and personal beliefs. Participants mentioned that their peers at school and within their communities identified with a wide variety of religions and having the availability of houses of worship within Auckland offered them a sense of comfort and belonging as they were able to practice their religion. However, places of worship were not only used for prayer, but it also was a place for finding a safe common ground, a place to socialize and be acknowledged. 

*I feel settled here, like in the religious side and with family I feel settled… like there are heaps of religious places you can go*.—*Female, 16 years old, Sri Lanka*

Most participants agreed there was no room for dwelling on the negative experiences and carrying on with a purpose was the best way forward. This was evident in their parents sacrificing their careers and lifestyle in order to start over in New Zealand. This sacrifice to migrate to New Zealand gave the young migrants a sense of both obligation and motivation to do well in school, help their parents with language barriers and make a financial contribution. 

*My parents worked really hard for me to come here and yeah... my dad was a doctor over there in India and over here he is unable to get a job as good as there, but he is struggling for us so that we can live here perfectly. It motivates me, ‘cause I think he works hard and I should do something for him*.—*Female, 16 years old, India*

The migration experience bought about new perspectives and self-reflection to some participants. Overall, it was common among the young migrants to explain that the challenges associated with migrating tend to ease over time while still requiring endurance and inner strength to manage the process. 

*I would say that I have very new experience of living with people with a different culture, different language. I’ve experienced the life of being independent…Well if someone is born here with their family so again it goes back to living independently, so again it might not have experiences life and experience like how to cope with loneliness or like... Missing all your friends… For me it’s like learning to cope with living a lonely life and how to find new friends but for them, it’s just like they’ve had friends from like when they can remember. So sometimes you have to cope with running low on gas*.—*Male, 19 years old, Iran*

### 3.4. Support Seeking and Overcoming Challenges

Participants spoke about their ability to cope by themselves and how they developed their self-reliance to overcome difficult situations. They were able to do so by finding comfort in the familiar things and keeping themselves occupied with activities that brought them happiness. In particular, activities that were at least similar to what they used to do in their former countries. For the young people interviewed in this study, finding comfort in the things that are familiar to their former life was important to their emotional wellbeing.

*Listen to some music, play games, make some new hobbies… I used to listen to a lot of music that used to give me confidence*.—*Male, 18 years old, Sri Lanka*

The majority of young people were unaware of support they could receive from external (school) networks. However, those that were aware of some external support services, were reluctant to engage with them. The main reason for this was because they felt uncomfortable talking to people they did not know. In addition, some participants felt that the language barrier held them back from expressing themselves or being fully immersed in an activity. 

*I never use outside services, I don’t know, I just feel weird to call and talk to someone random*.—*Female, 17 years old, Taiwan*

Social media was most commonly used with these young migrants to keep in contact with their friends and family overseas. Keeping these old relationships going made participants feel grounded and lifted in their moods.

*I do FaceTime with my friends in China, my home, something than can actually like fix me up… and push me up. That is the hardest part for most people who want to come here or already here*.—*Male, 16 years old, China*

Social media enabled young migrants to make new friends in New Zealand by acting as a mediator between cultures. These connections to were beneficial not only for friendship but also to help support academic performance by having informal conversations about schoolwork. 

*Because sometimes you’d be like ‘hey what’s your Instagram?’ and then chat about homework and stuff… Social media is really important for like connecting with friends and stuff*.—*Female, 17 years old, Taiwan*

Communication was heavily emphasized as something that will help other new migrants settle into the country. Speaking English was dependent on the ability to communicate and relate to others as well as a form of seeking support and friendships from others. 

*I would say talking is just really magical and helpful. If someone is hating themselves, I think it’s because of having small amount of friends. I think just talk to them and (find) ways to talk to them… I say you will get through any challenges (you are) facing… just be strong you will get through it*.—*Male, 16 years old, China*

## 4. Discussion

The young people interviewed in this study, although coming from diverse ethnic, cultural, and economic backgrounds, shared similar experiences of settling into New Zealand. Throughout the focus group discussions, young migrants expressed the key issues they faced while acculturating to New Zealand. Those issues were centered around changing family dynamics, negotiating two cultures, and facing language barriers. 

As young people tend to adapt to new environments and languages much faster than their parents or caregivers, they tended to take on additional responsibilities [21]. In this study, these additional responsibilities included assisting parents with financial help (e.g., through part-time work) to translating/interpreting when dealing with everyday issues such as banks, utilities companies and insurance providers on their parents’ behalf. These forms of informal assistance are often referred to as a form of ‘cultural or language brokering’ [22]. Cultural brokering is defined as the act of mediating between two cultures and language brokering is defined as the process of mediating information to caregivers, both concepts which are common across migrant families [23,24]. Although language and cultural brokering has several benefits on a young migrant such as instilling nurturing, supportive and caregiving behaviors in young migrants [25], it has also shown to be a cultural stressor for young migrants if their added responsibility was seen as a burden [25,26]. It is not clear whether the tendency toward taking on additional responsibility for elders is a factor related to migration or indeed an implicit cultural expectation (that young look after their elders). 

The young people in this study did not perceive the added responsibility placed on them as a burden, but rather expressed their appreciation of the opportunity to assume greater (adult) responsibility and therefore greater autonomy than their New Zealand born peers. Previous studies have attributed this increased level of autonomy to peer pressure in their host country [27]. However, the young migrants in this study indicated that they gained a sense of autonomy by facing challenges related to their acculturation journey such as taking on more responsibility in their families. Perhaps, for many, migration meant their parents sacrificing their homelands, careers, and lifestyles to seek a better life for their children. These ‘sacrifices’ contributed to the young migrants developing a sense of obligation to return the favor and assist their parents in various tasks. Although not directly explored, the theory of segmented assimilations is relevant to consider when understanding the several distinct trajectories that young migrants can follow. Portes and Zhou [28] proposed the theory of “segmented assimilation” as a way of describing different assimilation experiences such as classical assimilation, downward assimilation and selective acculturation. However, this study focuses on young migrant’s experiences of settlement into New Zealand rather than categorizing the outcomes of assimilation or exploring their socioeconomic status, social capital or family cohesion [29].

Participants emphasized that lacking English proficiency prevented them from participating in school, making friends and expressing themselves and consequently, feeling isolating. Escobar and Tamis-LeMonda [30] suggest that language skills can contribute to ones understanding of self, those around them and the environment, making it a primary resource for negotiating social identity. The special significance of friendships was repeatedly emphasized by the young people in this study. Although making friends was one of the most difficult obstacles they faced when they arrived in New Zealand, friendships were relied on particularly for emotional and academic support. For some, their ability to make friends was hindered by their English proficiency, and for others, negotiating two cultural worlds affected their ability to find the ‘right’ friends. This concept of needed social resources is evident in both developmental and acculturative research, indicating that having friendships is a factor that supports positive adolescent development as well as playing a role in the acculturation process [31].

The young migrants in this study drew upon a variety of strategies to cope with their migration related stress. They used their experiences of moving to New Zealand to display gratitude and self-growth. Most displayed an outward sense of independence and maturity as they expressed an understanding of their parents’ struggle to provide better opportunities in a new country and toward their ability to overcome the challenges they faced. Fundamentally, this feeling of gratitude seemed to be a factor that supported their ability to cope with acculturation stresses. The positive aspects of migrating overshadowed the more challenging times, making their perception of migration a worthwhile and fulfilling experience. Gratitude toward migration can be linked to research in areas of stress related growth and resiliency [32].

The young people in this study described their coping strategies to include things that were once familiar to them such as religion/spirituality and hobbies. This connection to the familiar may offer a sense of comfort, security, and a connection to identity. Pande [33] suggests migrants hold onto their traditional way of life as it forms a fundamental part of their earliest socialization. Religion/spirituality has been found to play an important role in acculturation process as it has been identified in the literature as having fewer internalizing and externalizing psychological symptoms [34]. In this study, some participants felt that New Zealand’s already existing multi-culturalism contributed to their settlement as they were able to easily find places of worship not only for prayer but for socialization with peers. 

The majority of young migrant interviewed in this study preferred coping with acculturation stressors privately. Young migrants in this study showed the tendency to retract inward relying on their own resources in managing stress, rather than seeking support. Despite the availability of support via school or external counselors or online tools, many were reluctant to use these due to their concerns about expressing personal feelings with someone they did not know. Although the participants appeared reluctant to use external services, they recommended it to others and acknowledged its effectiveness in certain situations. Factors such as language and cultural barriers may have also played a part in their decision not to seek help externally; informal help such as talking to a friend or close family member was preferred. Young people in this study were avid users of digital media, mainly using social media for both social connections and academic help. This acts as a “social glue” for transnational connections, meeting new friends and keeping in touch with former friends [35]. 

There are several limitations of this study that need to be acknowledged when interpreting the findings. The small sample size means this research only represents a relatively small proportion of young migrants’ experiences in three schools. The aim of the study was to explore the variety of pathways that migrant youth take during their settlement. By keeping the inclusion criteria open, it was felt that these experiences could be better represented and explored, giving a breadth and depth of information that could ultimately be beneficial for those in contact with young migrants of all ethnicities. However, creating a meta-story of all the participants’ stories, embedded into one, might have lost the uniqueness and diversity of the migrant youth. The findings from this study therefore capture only those experiences that were gathered during the interviews and cannot be extrapolated to other migrant experiences. We did not seek data saturation due to limited resources. Furthermore, as the interviewer had experienced migration to New Zealand as an adolescent, it is always possible that this connection may have introduced a social desirability responses to the interview questions. 

It is important to note that the research took place in Auckland, a city in which is unique compared to the rest of New Zealand. Auckland is recognized as one of the most multicultural cities of New Zealand (and one of the more diverse cities globally) and receives the highest number of migrants. Therefore, if this study were to be conducted elsewhere in New Zealand, findings may vary. We heard only a few negative experiences being described in this study, which may be due to the way our conversations were framed i.e., we purposefully asked young people to reflect on their whole experience and what helped them settle in the new home of New Zealand. 

The purpose of the study is to provide insight into the experiences and coping strategies used to overcome acculturation stress and the support that is needed at different points of their settlement. Throughout the process of developing and conducting this piece of research, it was found that several avenues remain to be explored. For instance, prospective studies offer a mechanism to follow-up a cohort of new migrant youth over time to build a more complex picture of their experiences and factors that exacerbate or mitigate these. In addition to this, one of the requirements to participate in this study was the ability to speak conversational English and it is important to acknowledge the limitations that this could bring forth. Young migrants with limited English proficiency are therefore likely to be underrepresented. In the future, exploring a wider range of English proficiency or having had translators present may have highlighted the demand for social support or cultural and language brokering within this group. 

It is possible also to examine the social network forming patterns among young migrants using social network analysis; this would provide again a dynamic visual representation of the social connectors and their characteristics. Both study designs would need to examine the role of social media in providing culturally and developmentally appropriate social support for young people who have recently undergone this major life transition. 

This research aimed to provide a platform for young migrants to voice their experiences of migration to New Zealand. These migration stories reflect some profound challenges that accompany the migration and acculturation process. For some this was demonstrated through a heightened sense of obligation to their families and having to take on adult responsibilities. Social policies and services dedicated to supporting migrant young people have been slower to recognize the specific needs of adolescents and young people. Although there was evidence that social media can play a supporting role in facilitating connections ‘back home’ it is also recognized as a double edged sword. Marlowe notes that these technologies (such as social media sites) can “enable and hinder possibilities for integration, translational connection and sites of belonging” [4]. 

How can services be better designed to support the challenging process of resettlement for adolescents? First, it is important that research privileges the lived experiences and coping strategies adopted by young migrants [36]. Schools play an important role in this process, but it is evident that some adolescents are adept at navigating their new social and academic environments with the aid of digital media. Despite the complex implications of heavy reliance on digital media, it has a potential role in providing a nexus for quality social, mental health and wellbeing services. Research into the perceived value and benefits of accessible, equitable and quality services (online and face to face) is also warranted.

## 5. Conclusions

This qualitative study gives voice to an often-overlooked population group. It provides insights that may be used to better understand the experiences of young migrants. Importantly, this research could be used to begin working on building support services for this vulnerable group. There may be value in extending in-person care with digital technologies. 

## Figures and Tables

**Table 1 ijerph-18-01292-t001:** Participant demographics across the six focus groups.

Focus Group No. and School Decile	Ethnicity	Gender	Age	Time in NZ	Who They Live with
1 (Decile 7)	Samoa	Female	17	3 years	Aunt and uncle
	China	Female	16	2 years	Mother and grandmother
	Sri Lanka	Female	17	3 years	Parents
	Taiwan	Female	17	1.5 years	Aunt and uncle
	India	Female	17	4 years	Aunt and elder brother
	China	Female	17	1 year	Parents
2 (Decile 7)	Iran	Male	19	2 years	Mother
	China	Male	16	3 years	Mother
	China	Male	17	3 years	Parents
3 (Decile 6)	Australia	Female	16	8 months	Grandmother
	Samoa	Female	18	6 months	Grandfather and elder brother
	Malaysia	Female	17	8 months	Parents
	Sri Lanka	Female	16	1.5 years	Parents
4 (Decile 4)	Sri Lanka	Female	19	4 years	Parents
	Sri Lanka	Female	17	3.5 years	Parents
	India	Female	17	4.5 years	Parents
	Cambodia	Female	17	7 months	Parents
	Pakistan	Female	16	4 years	Parents
	India	Female	16	1 year	Parents
	Malaysia	Female	16	2 years	Parents
5 (Decile 4)	Sri Lanka	Male	18	3 years	Parents
	Samoa	Male	17	3 years	Parents
	China	Male	16	5 months	Father
	Brazil	Male	17	1 year	Mother
6 (Decile 6)	China	Male	19	5 years	Parents
	Malaysia	Male	19	2 years	Parents
	Samoa	Male	16	1.5 years	Aunt and uncle
	Fiji	Male	17	5 years	Parents

## Data Availability

Data sharing is not applicable to this article as the data consists of focus group transcripts, which for reasons of confidentiality, cannot be shared.

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
