# Peer review of "“Just Be Strong, You Will Get through It” a Qualitative Study of Young Migrants’ Experiences of Settling in New Zealand"

_ijerph, 2021, doi:10.3390/ijerph18031292_

Round 1
Reviewer 1 Report
Thank you for forwarding this paper for review.
The paper is interesting and provides insight into settlement for young people, and the role of social work and other services in supporting them.
There does not seem to be a literature review as such, and the reader goes straight from the Introduction to the Methods – I don’t know if this is an error, or if that is a different style to one I’m used to.
There is a considerable literature internationally particularly in the UK, (see Kholi and others) and in Australia and New Zealand (see Jay Marlowe’s work – he’s based in Auckland) which might give you additional depth in your analysis. See for example these here:
Correa-Velez, I., Gifford, S.M. and Barnett, A.G., 2010. Longing to belong: Social inclusion and wellbeing among youth with refugee backgrounds in the first three years in Melbourne, Australia. Social science & medicine, 71(8), pp.1399-1408.
Gifford, S.M., Bakopanos, C., Kaplan, I. and Correa-Velez, I., 2007. Meaning or measurement? Researching the social contexts of health and settlement among newly-arrived refugee youth in Melbourne, Australia. Journal of refugee studies, 20(3), pp.414-440.
Kohli, R.K. and Mitchell, F. eds., 2007. Working with unaccompanied asylum seeking children: issues for policy and practice. Macmillan International Higher Education.Marlowe, J., 2017. Belonging and transnational refugee settlement: Unsettling the everyday and the extraordinary. Routledge.
The Discussion brings together some rich themes, and links well to the literature. This may be the focus you intended (given my comment above). I would like to see more critical engagement with the literature and that this is linked to broader social policy and settlement issues. One could argue this approach has similar features to many countries, and it would be interesting to link this to broader policy and research agendas. If this could be linked further to some of these overarching debates, it might have more interest and relevance to an international audience.
There were a few minor typos throughout, i.e. line 176– so please carefully check this again.
Please don't be discouraged by making additional changes to improve this work - it will really strengthen what is already a good paper with rich data.
Best wishes and good luck with your future work.
Author Response
Thank you for forwarding this paper for review.
The paper is interesting and provides insight into settlement for young people, and the role of social work and other services in supporting them.
There does not seem to be a literature review as such, and the reader goes straight from the Introduction to the Methods – I don’t know if this is an error, or if that is a different style to one I’m used to.
There is a considerable literature internationally particularly in the UK, (see Kholi and others) and in Australia and New Zealand (see Jay Marlowe’s work – he’s based in Auckland) which might give you additional depth in your analysis. See for example these here:
Correa-Velez, I., Gifford, S.M. and Barnett, A.G., 2010. Longing to belong: Social inclusion and wellbeing among youth with refugee backgrounds in the first three years in Melbourne, Australia. Social science & medicine, 71(8), pp.1399-1408.
Gifford, S.M., Bakopanos, C., Kaplan, I. and Correa-Velez, I., 2007. Meaning or measurement? Researching the social contexts of health and settlement among newly-arrived refugee youth in Melbourne, Australia. Journal of refugee studies, 20(3), pp.414-440.
Kohli, R.K. and Mitchell, F. eds., 2007. Working with unaccompanied asylum seeking children: issues for policy and practice. Macmillan International Higher Education.
Marlowe, J., 2017. Belonging and transnational refugee settlement: Unsettling the everyday and the extraordinary. Routledge.
Response: Thank you. We acknowledge that we have prepared tightly edited introduction, not a full literature review. We have added a further statement to reference the suggested publications and agree that this strengthens the link to recent and relevant literature.
The Discussion brings together some rich themes, and links well to the literature. This may be the focus you intended (given my comment above). I would like to see more critical engagement with the literature and that this is linked to broader social policy and settlement issues. One could argue this approach has similar features to many countries, and it would be interesting to link this to broader policy and research agendas. If this could be linked further to some of these overarching debates, it might have more interest and relevance to an international audience.
Response: The reviewer makes some very useful comments here and we have taken these onboard. We have added some text to address the implications of our work to the broader social policy and recommendations for practice (page 10).
There were a few minor typos throughout, i.e. line 176– so please carefully check this again.
Response: Thank you, we have proofed our manuscript thoroughly.
Or please see the attachment

Reviewer 2 Report
The article concerns the issue of adolescent migration in New Zeland. The general organization of the article is clear and the definition of objectives, methods and results are well organized. However, it may be useful to add in the methods section the contents of the semi-structured interview proposed to the focus groups and to specify who conducted the focus groups, how many focus groups were held. Both of these aspects can have an impact on the results: for example, the fact that the interviewees did not identify major integration difficulties could be connected to a social desirability bias if the interviewers were indigenous people. Anyway the most relevant aspect to be reviewed concerns the discussion of the results: the authors can appropriately integrate how the interviewees’ former cultural factors can impact on the their spechees. For example, is the adolescents’s sense of obligation to their parents and taking of their adult responsibilities only due to the migration event or is it also linked to the former culture? In the same perspective the protective effect of having goals and objectives to be achieved can be connected both to the migration challenges and to the culture of origin of the focus group participants. For these reasons it is necessary to discuss the results of the research also in the cultural perspective of the participants.
Author Response
The article concerns the issue of adolescent migration in New Zeland. The general organization of the article is clear and the definition of objectives, methods and results are well organized. However, it may be useful to add in the methods section the contents of the semi-structured interview proposed to the focus groups and to specify who conducted the focus groups, how many focus groups were held. Both of these aspects can have an impact on the results: for example, the fact that the interviewees did not identify major integration difficulties could be connected to a social desirability bias if the interviewers were indigenous people.
Response: We have included the list of key topics covered in the focus group interviews (page 3).
We have also added detail on the facilitator of the interviews and noted in the limitations section that this may have affected the narratives collected. (page 9)
Anyway the most relevant aspect to be reviewed concerns the discussion of the results: the authors can appropriately integrate how the interviewees’ former cultural factors can impact on the their spechees. For example, is the adolescents’s sense of obligation to their parents and taking of their adult responsibilities only due to the migration event or is it also linked to the former culture? In the same perspective the protective effect of having goals and objectives to be achieved can be connected both to the migration challenges and to the culture of origin of the focus group participants. For these reasons it is necessary to discuss the results of the research also in the cultural perspective of the participants.
Response: This is a good point; we have added further detail to the discussion on the impact of ‘formative’ cultural norms and expectations that carry over to the new cultural environment (page 8).
Or please see the attachment.

Reviewer 3 Report
1. There are not enough relevant suggestions for this research and the practical significance of this article is not sufficiently highlighted.
2. The sample size of this study is relatively small and it is not universal to select samples from only three schools in Auckland. It is recommended to investigate more data from other regions for comparative analysis.
Author Response
- There are not enough relevant suggestions for this research and the practical significance of this article is not sufficiently highlighted.
- The sample size of this study is relatively small and it is not universal to select samples from only three schools in Auckland. It is recommended to investigate more data from other regions for comparative analysis.
Response: Thank you. We have noted the small sample size in the limitations. However, consistent with this methodology, we were seeking richness from the data not breadth or representativeness (page 9).
Or please see the attachment.
